# Introducing Dynamic Object Creation to PDDL Planning

**Stefan Edelkamp**
King's College London, UK
edelkamp@kcl.ac.uk

**Alberto Lluch-Lafuente**
DTU, Copenhagen, Denmark
albl@dtu.dk

**Ionut Moraru**
King's College London, UK
ionut.moraru@kcl.ac.uk

## Abstract

For automated planning robot behaviour in a rather unexplored world, often there are exogenous events introducing new objects triggered by the actions that were unforeseen in the initial state of the planning domain model. This suggests extending PDDL, on every of its language levels, by a feature for dynamic object creation. Syntactically, this is dealt with by adding a keyword *new* to PDDL. Semantically, the core challenge in dealing with dynamically created objects is that they lead to a growing state representation during the plan space exploration. As almost all existing automated planners are not designed to deal with varying state vectors —mainly due to relying on grounding of the lifted PDDL input in their static analysis stage— in this short paper we present a first solution to solve the dynamic object creation problem via a translation of PDDL model into a term rewriting tool, where plans can be found.

## Introduction

Unforeseen things happen to all of us every day. It is widely seen as a matter of intelligence for any agent and especially us humans to be able to adapt to situations that are new.

In machine learning this requirement of intelligent behavior is referred to as a problem of generalization and has let to distinguish between training and test sets (Flach 2012).

In exploratory robotics, however, like a rover operating on mars, often we have unexpected exogenous events triggered by controlled or uncontrolled actions (Fox, Howey, and Long 2005). In terms of action planning, new objects may pop up (and disappear) during executing a plan.

By introducing the keyword *new* to the planning domain definition language PDDL (Fox and Long 2003), domain objects might be created and all according predicates and functions initialized. One apparent option is to use the *new* operator as an action effect. Quantification can then be used to initialize the predicates and functions for coping with the new object. Together with this initialization of variables, the requested PDDL language extension for dynamic object creation are considerably small.

While in software model checking and graph transformation newly created objects like processes or graph nodes/edges are common (Artho and Visser 2019; Holzmann 2004), current planning technology is limited in the ability to deal with dynamic objects. While syntactically rather easy, introducing them to PDDL domains will lead to varying state vectors and substantial internal changes of planners and static analyzers (Helmert 2006). At least conceptually, for explicit-state forward-chaining planning, a varying state representations can be made available.

There are other extensions to PDDL that are more of syntactical nature, and lead to compilation schemes (Nebel 1999). For example object arrays and records could be compiled away much in the spirit to what can be done with ADL types and negated preconditions. Compiling conditional effects may lead to an exponential blow-up in the planner input size (Pednault 1989).

In computational complexity terms, the main problem of dynamic objects, besides proper duplicate detection, is a potential infinite state space.

This position paper is a feasibility study suggesting that PDDL should allow actions that create new objects. It provides in a small case study to illustrate that via a compilation to a term rewriting input and by calling a rewriting tool like Maude (Clavel et al. 2007), planning with dynamically created objects is indeed possible. Possible alternative encoding of domains for software model checkers or graph transformation systems are briefly discussed.

## Problem Statement

For the dynamic creation of domain objects the syntactical modifications to PDDL (Fox and Long 2003) are small and rather straight-forward. Beside ordinary action effects, we additionally allow object creation with another effect of the following syntax:

```
(new (a - <type>) (:init (<formula>*)),
```

where `a` is the new object and `name` would be the new object type. As with the initial state, the closed world assumption in PDDL suggests that all predicates involving `a`, not explicitly mentioned in the formula for `:init` are false. A simple example would be a new block appearing on the table in the Blocksworld domain (Slaney and Thiébaux 2001).

Similarly, we can define a *delete* operator for PDDL planning, which is aimed deleting an object from the planning problem, such as the top block of a Blocksworld tower.

```
(delete (a - <type>))
```

For this case, all predicates and functions concerning `a`, would be removed from the current state description. We briefly look at if this PDDL language extensions is essential.

**Proposition 1 (Complexity of Dynamic Object Creation)**
*STRIPS planning with dynamic object creation is undecidable.*

The rough argument is as follows. Based on the known STRIPS encoding of Bylander (1994) of a Turing machine for proving the PSPACE-completeness of STRIPS, it is easy to deduce that with newly created objects, even STRIPS planning, becomes undecidable, as we may encode the working of a Turing machine with an infinite tape and cells being created on the fly. Undecidability then directly follows from the Halting problem.

As with the undecidability result on planning with numbers this does not say anything about the fragment of benchmark planning problems looked at. Moreover, as in every semi-decidable setting, one can always be lucky in finding a plan, in case exists.

### Dynamic Blocksworld

We next consider dynamic object creation in Blocksworld, where new blocks can be added to the domain description using additional actions containing the *new* operator. The following example of a dynamic Blocksworld domain allows new blocks to be introduced to the planning state in two different ways.

As there are books on the topic (PDD ), we skip a introduction to PDDL. We also avoid a formal treatment of a planning task. For the interested reader not from the planning community it is suffcient to know that we mainly assume operators that consist of a precondition and add and delete lists,

Figure 1 provides a possible PDDL description with two actions containing the *new* operator.

Most recent planners ground the PDDL inputs, instantiating the proposition, function, and action parameters to the objects. While there are different planners that do not necessarily ground the lifted input domain, like TLPLAN (Bacchus and Kabanza) or version of POPF (Benton, Coles, and Coles 2012), at least in competitive planning, the problem of dynamic objects has neglected for a long time. There were simply no benchmarks available.

### Settlers

The problem of dynamic object goes back to the introduction of PDDL2.1 (Fox and Long 2003), and before. In these planning benchmarks workarounds have been found for the apparent need to create new objects. One example is the *Settlers* Domain as proposed for the International Planning Competition 2002[1]. One critical action for constructing a car is shown in Fig. 2. There are similar ones for constructing trains and ships. The critical aspect for these domain is that the number of potential vehicles is fixed in the problem description and only the type of the object as well as its initial load is fixed in the action effect.

The Settlers domain was designed for the numeric track, and proved to be a very tough resource management domain. Several interesting issues in encoding arise as well

[1] https://github.com/potassco/pddl-instances/blob/master/ipc-2002/domains/settlers-numeric-automatic/domain.pddl

```
(define (domain blocksworld)
(:requirements :strips :typing)
(:predicates (clear ?x - block)
             (on-table ?x - block)
             (arm-empty)
             (holding ?x - block)
             (on ?x - block ?y - block))
(:action pickup
  :parameters (?x - block)
  :precondition
   (and (clear ?x) (on-table ?x) (arm-empty))
  :effect (and (holding ?x) (not (clear ?x))
  (not (on-table ?x)) (not (arm-empty))))
(:action putdown
  :parameters  (?x - block)
  :precondition (and (holding ?x))
  :effect (and (clear ?x) (arm-empty)
  (on-table ?x) (not (holding ?x))))
(:action stack
  :parameters  (?x ?y - block)
  :precondition (and  (clear ?y) (holding ?x))
  :effect (and (arm-empty) (clear ?x) (on ?x ?y)
    (not (clear ?y)) (not (holding ?x))))
(:action pop-up
  :parameters
  :precondition ()
  :effect
    (new (?x - block)
      (:init (on-table ?x) (clear ?x))))
(:action pick-new
  :parameters
  :precondition (and (arm-empty))
  :effect
    (new (?x - block)
      (:init (holding ?x))))
(:action unstack
  :parameters  (?x - block ?y - block)
  :precondition
  (and (on ?x ?y) (clear ?x) (arm-empty))
  :effect (and (holding ?ob) (clear ?y)
    (not (on ?x ?y)) (not (clear ?x))
    (not (arm-empty)))))
```

Figure 1: Blocksworld domain with create-block action.

as the subsequent problem of planning with the domain. In particular, resources can be combined to construct vehicles of various kinds. Since these vehicles are not available initially, this is an example of a problem in which new objects are created. PDDL does not conveniently support this concept at present, so it is necessary to name potential vehicles at the outset, which can be realized through construction. A very high degree of redundant symmetry exists between these potential vehicles, since it does not matter which vehicle names are actually used for the vehicles that are realized in a plan. Planners that begin by grounding all actions can be swamped by the large numbers of potential actions involving these potential vehicles, which could be realized as one of several different types of actual vehicles.

```
(:action build-cart
  :parameters (?p - place ?v - vehicle)
  :precondition
      (and (>= (available timber ?p) 1)
           (potential ?v))
  :effect
   (and
     (decrease (available timber ?p) 1)
     (is-at ?v ?p)
     (is-cart ?v)
     (not (potential ?v))
     (assign (space-in ?v) 1)
     (forall (?r - resource)
       (and (assign (available ?r ?v) 0)))
     (increase (labour) 1)
   )
)
```

Figure 2: Action in settlers domain with create-block action.

## Limits and Possibilities of Compilation Schemes

As it is certainly desirable to have planners that can deal with the new language feature, one would like existing planners to solve transformed benchmark domains without it.

Such a transformation to ordinary PDDL is possible, if objects are not deleted (as it is in the two examples above), and if one has an upper bound on all possible objects to be generated available. In general, however, as shown in the theorem, finding such a super set is as difficult as the plan existence problem, so that we cannot expect a fully automated compiling scheme without the problem designer's help.

We briefly sketch some transformation details to clarify the semantics of the language extension. Through this will not be formal construction, it indicates necessary changes.

The transformation uses some special predicates as flags to govern the appropriate object handling. Most importantly we include a predicate `(is-created ?a - <name>)` in the domain description, and precondition every parameter object in every action, if it is already created.

An action with the effect `(new (a - <name>)` `(:init (<formula>*))` is transformed introducing an ordinary add-effect `(is-created ?a)`. Since ?a is not yet a parameter of the action the compilation will include an extra parameter into the transformed action.

For example, consider action `create-block` that has no parameter and the only effect is `(new (a - block) (:init (holding ?a)))`. It will be transferred to an action `create-block (?a - block)` having `(holding ?a)` in the effect list. Certainly, the translated action should require `(not (is-created ?a))` included in the precondition list. (Actually it would only be necessary to include the additional predicate `not-is-created` into the precondition list and to guarantee that in each action the predicates `not-is-created` and `is-created` are complement to each other.)

## Term Rewriting for Planning with Dynamic Object Creation

A term rewriting rule is a pair of terms to indicate that the left-hand side can be replaced by the right-hand side. A term rewriting system is a set of such rules.

Roughly speaking, term rewriting for STRIPS problems with operators $o = (P, A, D)$ is as follows. We have rules of the form

$$LHS => RHS$$

with

$$RHS = A \cup (P \setminus D), \text{ and} \quad (1)$$
$$LHS = P \cup D \quad (2)$$

As often made assumption is $D \subseteq P$ we have $LHS = P$. Formally, it is not correct that PDDL itself *assumes* $D \subseteq P$", and also not if PDDL is replaced with STRIPS. However, one might assume this setting in a compilation to transition normal form (Pommerening and Helmert 2015).

As we have preconditions in action planning, instead of going for only ordinary rewriting, we actually require conditional rewriting.

Deletion in such conditional rewriting systems is realized by some facts on the rule's LHS not to be present in the rules RHS. To do deletion in the semantically correct way, however, one has to be careful. In graph rewriting there are various semantics: one allows to remove any object (and then any reference to that object must be garbage collected), another one forces you capture all existing references to the object to be deleted in the rule's LHS. In both cases the idea is to avoid *dangling edges* (edges pointing to a node/object that does not exist anymore).

The fortunate thing about term rewriting is that one can deal with lifted problem representations, so that there is an easy translation of PDDL models that contain object creations and deletions. One example is the Blocksworld Maude model in Figure 3. Maude (Clavel et al. 2007) is essentially term rewriting and therefore objects, their creation and deletions are not first class features. Fortunately, it is very easy to encode them (as done in many applications of Maude) for instance mimicking graph rewriting theories (which do have creation and deletion as first class features).

The term rewrite is started with the initial and goal state specification

```
Maude> search empty & clear(c) & clear(b)
       & table(a) & table(b) & on(c,a)
       =>* empty & clear(a) & table(c) &
       on(a,b) & on(b,c)
```

finds the correct plan without object creation. For the call

```
Maude> search [1] empty & clear('c) & clear('b)
 & table('a) & table('b) & on('c,'a) & next(0)
=>*
empty & clear(0) & table('c) & on('a,'b) &
on('b,'c) & on(0,'a) & next(x:Nat)
```

finds the correct plan with objects being created:

```
    unstack putdown pickup stack
    pickup stack create pickup stack
```

```
mod BLOCKS-WORLD is
 protecting QID .
 sorts BlockId Prop State .
 subsort Qid < BlockId .
 subsort Prop < State .

 op on-table : BlockId -> Prop .
 op on : BlockId BlockId -> Prop .
 op clear : BlockId -> Prop .
 op holding : BlockId -> Prop .
 op empty : -> Prop .
 op 1 : -> State .
 op _&_ : State State -> State [assoc comm id: 1]
 vars X Y : BlockId .

 rl [pickup] : empty & clear(X) & on-table(X)
  => holding(X) .
 rl [putdown] : hold(X)
  => empty & clear(X) & on-table(X) .
 rl [unstack] : empty & clear(X) & on(X,Y)
  => holding(X) & clear(Y) .
 rl [stack] : holding(X) & clear(Y)
  => empty & clear(X) & on(X,Y) .
 --- Natural numbers as id's as well
 pr NAT .
 subsort Nat < BlockId .
 --- New operator for the next free id
 op next : Nat -> Prop .
 --- Variable for states and naturals
 vars S : State .
 vars n : Nat .
 --- Rule for creating blocks
 rl [create] : next(n)
  => table(n) & clear(n) & next(s(n)).
 --- Rules for deletion of anonymous roofs
 rl [delete1] : table(n) & clear(n) => 1 .
 rl [delete2] : on(n,X) & clear(n) => clear(X) .
endm
```

Figure 3: Dynamic Blocksworld in Maude.

The solver does not do any optimization nor heuristic search and simply reports the first solution obtained.

The drawbacks of creation/deletion not being first class features means that the tool will not take care of name reuse which is essential to manage such infinite state spaces (which can be reduced to finite ones if the number of objects per state is bounded).

While it is true that the main engine performs breadth-first search, there has been some efforts to develop a strategy language to build smart search strategies on top of Maude.

The main idea is that Maude treats specifications (modules) as ordinary data. There is a module in Maude called META-LEVEL that contains sorts (i.e. types) for modules, equations, rules and all that. In addition it contains functions to simulate rule application at the meta-level or to perform a search (at the meta-level). Moreover, since META-LEVEL is a module, we can also have meta-meta-representation, and meta-meta-meta-representation and so on. In practice with this reflection mechanism we can do meta-programming. For instance we can define some re-factoring (like optimiz-

ing Maude modules), translations (e.g. from signature to another), or analysis tools (e.g. the original LTL model checker was implemented in Maude itself), etc. This has the advantage that one uses the same language (Maude) both for the tools and for the specifications and that, since everything developed is a couple of Maude modules one can apply new tools both for your specifications and to the tools.

Note that the naive model can be improved by applying some tricks for name re-use (for instance once we delete an object $n$ we could decrease the counter and the $id$ of all objects with id greater than $n$). In that case if one allows creation up to a certain bound (e.g. the maximal number of blocks allowed in the game) then we are finite state.

Maude rules can have conditions, but if one want negative application conditions like *there is no other block on the table* which are essentially *global* we have to change the rules of the example a bit since they are essentially *local*. One would need something like an operator for enclosing states [ _ ] and rules in the following style where a variable $S$ is used to capture *the rest of the propositions forming the state*.

```
crl [ S & my-preconditions]
 => [S & my-effects]
```

Recall that in Maude we can also have rewrite steps in the conditions (and even use their result), which turns out to be a very expressive mechanism.

## Discussion on Semantic Canveats

The example where a single object appears without any dependence on existing or simultaneously created other objects.

Think about a Mars rover, where objects of interest (of type object-of-interest) can be detected whenever a traverse action was executed. Each object of interest has an assigned location (of type location), which is modeled with the predicate `object-at(object-of-interest, location)`. A traverse action between 2 locations to an action where both an object-of-interest and a location are created such that the created object of interest is at a newly created location could look as follows.

```
(:action traverse
  :parameters (?l1 ?l2 - location)
  :precondition (rover-at ?l1)
  : effect (and
        (not (rover-at ?l1))
        (rover-at ?l2)
     (new (?o - object-of-interest
          ?l - location)
        (:init (object-at ?o ?l))))
)
```

If objects are added, there must be some clever mechanism to set them into relation with other objects that have already existed. One possibility could be *conditional initializations*. Take, for instance a planning task with a predicate `(pred ?o1 ?o2 - obj)` that should hold for a newly created object ?a of type `obj` and all existing objects ?b of type `obj` if and only if another predicate `(cond ?o - obj)` holds for ?b. One could use the same syntax that is used for conditional effects, i.e.,

```
(new (?a - object)
  (:init (forall (?b - object)
    (when (cond ?b) (pred ?a ?b)))))
```

The introduction of *new* and *delete* will lead to states of variable size, but also to variable size of ground actions. This is a challenge on the implementation side as concepts like invariant synthesis or reachability analysis are significantly harder if the number and form of actions is unclear.

Of course there is an immediate influence of added and deleted objects on the goal (and other conditions that want to refer to the objects). A question that immediately comes to mind is what happens in an instance with:

```
(:objects o1 o2 o3 - my-object-type)
(:goal (and
  achieved(o1)
  achieved(o2)
  achieved(o3)))
```

where the predicate "achieved" must be true for all (initially existing) objects o1, o2 and o3. What happens now if o1 is deleted? The intuitive answer would be that it is no longer possible possible to reach a goal state, because the object is explicitly mentioned in the goal and it's impossible to achieve that goal once o1 has disappeared. However, what happens if achieved(o1) was true in the moment that o1 disappeared? Furthermore, what happens if the goal is given via a quantifier as

```
(:objects o1 o2 o3 - my-object-type)
(:goal (forall (?o - my-object-type)
        (achieved(?o))))
```

which is equivalent under the current PDDL semantics. Intuitively, we would assume now that removing o1 is not a dead end as it is still possible to achieve the goal for all existing objects since o1 is not mentioned explicitly, but then the two goals are no longer equivalent. Similarly, what happens in the second case if another object of type my-object-type is created.

The mapping of PDDL to Maude is not perfect. There are hacks needed to make conditional rewriting work like next(x:Nat) in the example, which certainly is far beyond the LHS => RHS rules.

Of course breadth-first search should also be able to provide a solution to this kind of planning tasks and is a concept that is universally understood. The need of using a rewriting tool like Maude is needed as most planners suppose grounding prior to the search and am not capable to deal with objects that dynamically created.

## Further Opportunities for Planning with Dynamic Objects

Our interest is in extending the planning language to include the concept of object creation. We are presenting Maude as one example to show that such extended problems can be solves, but we are not arguing that is the only way to solve these problems.

In fact, there is much related work in different research areas, of which we only provide the tip of the iceberg. We found one pioneering research in the area action planning.

Besides conditional term rewriting there is a larger body of research areas which could also be consulted as a basis for PDDL planning with dynamic object creation. We briefly will look into model checking and graph transformation.

## Action Planning

There is one related publication in the area of planning that compiles irrelevant objects to counters with special case of creation planning (Fuentetaja and de la Rosa 2016). The authors discuss how to deal with cases where objects are created in PDDL. They do not use a new operator, but specialized predicates. Later on, they discussed adding a keyword to the action description :newobjects in between :precondition and :effect. They used a intuitive motivating example called the *pizza domain*, where one piece is cut into two.

```
(:action cut
:parameters (?slice ?s1 ?s2 - slice
            ?t - tray ?z ?zhalf - size)
:precondition (and (not (= ?s1 ?s2))
  (holding ?slice ?t)
  (pizzasize ?slice ?z)
  (nextsize ?z ?zhalf)
  (notexist ?s1) (notexist ?s2))
:effect (and (freearms)
  (not (holding ?slice ?t))
  (not (notexist ?s1)) (not (notexist ?s2))
  (ontray ?s1 ?t) (pizzasize ?s1 ?zhalf)
  (ontray ?s2 ?t) (pizzasize ?s2 ?zhalf)))
```

Given a slice, the action *cuts* it into two new slices of half size. Each new slice is represented as a domain object and identified by a constant. The special predicate *notexist* represents that the corresponding slice identifier has not been used before the application of the action.

Their compilation to counters is insightful and involved using rewriting of the domain (including involved actions, initial and goal state) via a translation of the (lifted) PDDL input into one with numbers, eventually serving a compiled planning problem numeric planner to solve. For the action above this looked as follows.

```
(:action cut1
:parameters
  (?zhalf - size ?z1 - size ?t - tray)
:precondition (and
  (>=(Nslice-pizzasize_holding ?z1 ?t) 1)
  (>=(Nslice-notexist) 2))
:effect (and (freearms)
  (decrease (Nslice-notexist) 2)
  (decrease
    (Nslice-pizzasize_holding ?z1 ?t) 1)
  (increase
    (Nslice-pizzasize_ontray ?zhalf ?t) 2)))
```

The authors also used some modification to recent IPC benchmarks like *Gripper*, or *Childsnack*, and introduced some new ones like*Spanner*, and *Carpenter*. The work relates to known compilations of functional STRIPS (Geffner 2000).

## Model Checking

Model checking was introduced as an automatic verification technique for finite state concurrent systems (Clarke, Grumberg, and Peled 1999). Specifications are written in propositional temporal logic. The verification procedure is an exhaustive search of the state space of the design. There is a tight connection and exchange of techniques from and to the area of action planning.

Software model checkers of formal software models like SPIN (Holzmann 2004) with its input language *Promela* go one step further and allow concurrently running processes to be created using the *run* operator. This way the state vector is actually dynamic, which poses some advanced hashing and storing strategies. An example is leader election. It is not difficult to alter it to (Dynamic) Blocksworld.

```
byte leader = 0;
proctype N(chan chin, chout) {
 byte tok; chout ! _pid;
 do
 ::  chin ? tok ->
   if
   :: tok <  _pid -> skip
   :: tok >  _pid -> chout ! tok
   :: tok == _pid -> leader = leader+1; break
   fi
 od
}
init {
 bool f[5] = true;
 chan ab = [1] of {byte}, bc = [1] of {byte};
 chan cd = [1] of {byte}, de = [1] of {byte};
 chan ea = [1] of {byte};
 do
 :: f[0]-> atomic { run N(ea,ab); f[0]=false }
 :: f[1]-> atomic { run N(ab,bc); f[1]=false }
 :: f[2]-> atomic { run N(bc,cd); f[2]=false }
 :: f[3]-> atomic { run N(cd,de); f[3]=false }
 :: f[4]-> atomic { run N(de,ea); f[4]=false }
 :: else-> break
 od
}
```

While general functionality is granted, many benchmarks can be re-coded as active processes. When checking real software tools like the Java Path Finder (Artho and Visser 2019) and Steam (Leven, Mehler, and Edelkamp 2004) represent the changes to the memory pool for each system state.

## Automated Theorem Proving

Automated Theorem Proving is an area of study to get computers to prove logical and mathematical statements not just enumerating instances of a theorem exhaustively, but applying logical deduction, induction, inference and search strategies (depth first, breadth first, best first, iterative deepening) to arrive at a solution.

The main application is the formalization of mathematical proofs and in particular formal verification, which includes proving the correctness of computer hardware or software and proving properties of computer languages and protocols. The most widespread instance of Isabelle nowadays is Isabelle/HOL (Nipkow, Paulson, and Wenzel 2002), which

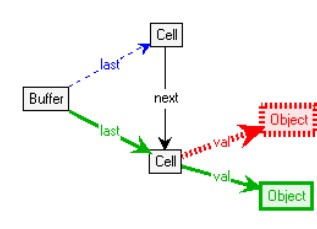

Figure 4: Graph transformation rule for object creation

provides a higher-order logic theorem proving environment with general proving mechanism called auto tactics, which should be tried before running external provers using the *sledgehammer*.

Isabelle/HOL includes powerful specification tools, e.g. for data types, inductive definitions and recursive functions with complex pattern matching. A for natural numbers, there is a proof principle of induction for infinite lists of yet unkown object. Induction over a list is essentially induction over the length of the list, although the length remains implicit. It might be possible to convert a PDDL model with object creation to an input of Isabelle.

## Graph Transformation

Software engineers are used to deal with annotated graphs. These models can be described by means of suitable graph transformation systems (Heckel 2006). In many cases, modeling must be complemented with analysis capabilities to understand how designed transformations behave and whether requested requirements are fulfilled. Powerful solutions bypass this limitation and allow the user to reason on how the different rules impact the behavior of the graph as a whole.

In graph transformation with tools like Groove (Zambon and Rensink 2018), there are rules that add and delete nodes and edges to the existing graph, through a process that is called *pushout*, which leads to the creation and deletion of objects. A simple example is shown in Fig. 4. There is interconnection with planning. Planning for graph transformation is discussed by (Edelkamp and Rensink 2007) and heuristic search planning for graph transformation systems by (Edelkamp, Jabbar, and Lluch-Lafuente 2006).

## Conclusion and Discussion

This paper proposes an extension of PDDL that allows to dynamically create and destroy objects and proposes this as a basis for running a planner competition in the future. The IPC has traditionally been closely related to extensions of PDDL. Perhaps, future editions of the IPC could consider a special track with this extensions even. As it seems hard to get current planners to support such an extension of PDDL, in the paper we also discussed different avenues and technologies to actually solve such extended planning tasks.

AI and action planning are both concerned about mastering the unknown. Therefore, object creation is both an important extension to PDDL modeling, as well as fascinat-

ing topic for research and planner development. With this position paper, we propose the keywords *new* and *delete* to PDDL.

We showed that conditional term rewriting is one adequate solution to the problem. Mapping PDDL to Maude input and extending it for creating new objects by means of rewriting PDDL directly in Maude is feasible, as it has been already done for many languages (including Java, C, process algebras, etc.). We have seen a first proof-of-concept, in which we played with simple, though interesting examples, to see what is gained from using Maude. New insights for PDDL or planners, and yet another planner are possible. Term rewriting can also be tool for solving additional (not only plan scheduling) problems of planning specifications like confluence checks to see which actions can be safely executed concurrently and which not. Planning with uncertainty may be available via logic programming and narrowing in Maude.

One of Maude's advantages is the fact that the language is reflective, which enables powerful meta-programming applications and the re-use of techniques. Investigating the issue of strategies/heuristics for Maude could, on the other hand, be the contribution of the planning community to Maude. There have been some efforts towards the efficiency of rewriting in the form of competitions, but we not aware on solving reachability problems with heuristics.

**Acknowledgement**   We are indebted to the brilliant reviews on this paper, highlighting that while the syntactic features for introducing the concept of object creation are rather simple, before running a track in the IPC the semantic consequences of such PDDL language extension has to be thought of in some larger panel.

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
