# OpenReview forum: "Introducing Dynamic Object Creation to PDDL Planning"
_icaps-conference.org/ICAPS/2019/Workshop/WIPC_

### Official Review · AnonReviewer3 · 2019-04-25
**Interesting new extension to PDDL**

**Rating:** 7
**Confidence:** 3

**Review:**

This paper proposes an extension of PDDL that allows to dynamically create and destroy
objects. The IPC has traditionally been closely related to extensions of PDDL, so it makes
sense to discuss this extension in the workshop. Perhaps, future editions of the IPC could
consider a special track with this or other extensions even though it seems hard to get
current planners to support such an extension of PDDL.

Some comments on this part of the paper:

 - One issue with the extension of PDDL is that it does not discuss how the newly created
   objects can be referenced in the goals of the problem. For example, if I specify a task
   where I have to create objects with certain properties, the goal should be some
   existentially (or universally) quantified formula over the objects in the state.

 - The dynamic blocksworld example looks a bit artificial to me (blocks are not so easily
   created and destroyed), and is not clear why pop-up or pick-new would be relevant.  How
   can one specify a meaningful goal in this domain that involves newly created blocks?

 - There is one related publication that should be discussed. In "Raquel Fuentetaja, Tomás
   de la Rosa: Compiling irrelevant objects to counters. Special case of creation
   planning. AI Commun. 29(3): 435-467 (2016)", the auhors discuss how to deal with cases
   where objects are created in PDDL. This porbably should be discussed in the compilation
   section, and perhaps could also be used as motivating example in the introduction (note
   that they introduce some recent IPC benchmarks like childsnack) .


After explaining the extension of PDDL the paper discusses how to do planning in this
setting, describing related approaches from model-checking. The two parts of the paper
seem a bit disconnected, and there is a "gap" between the section on compilation, to the
section "Term Rewritting". I suggest to have some introductory text explaining that it is
not clear how to do planning with the new extension and that this may be related to other
approaches in model-checking such as Maude that use term rewritting. I'd also change the
title "Term Rewritting" to something like "Planning with Dynamic Object Creation".

The explanation of the model-checking part is hard to follow for the workshop audience,
who is not so familiar with model-checking syntax. In particular, the models of Figure 3,
and in the model checking section (figure 4), should be explained in a bit more of
detail. For example, it is not clear to me how to recover a complete plan, from the one
that Maude finds. We know that a unstack action must be applied first but we do not know
which block should be unstacked.

One thing that is also not clear is whether the paper is presenting Maude as an example or
arguing that is the way to go. Some things suggest that such a claim is being made:
 - The description of Maude includes some parts describing all features of Maude, some of
    which do not seem related to the proposed PDDL extension. What the META-level has to
    do with object creation?

  - Why separate Maude in one section and then present other alternatives in a "related
    work" section? To me it would be more natural to have all these approaches at the same
    level, and discuss them as alternatives on how to approach planning with dynamic
    object creation. Otherwise, it should be more clear why the authors think that Maude
    is particularly interesting for modelling this type of planning problems.



Minor comments:
 - abstract: leads -> lead
 - p1. may leads -> lead
 - p1. if the PDDL language extensions is essential -> this PDDL language extension
 - p2. or version of POPF -> "a version", but which version?
 - p3. There is module META-LEVEL
 - p4. meta-meta-representation -> meta-meta-representations

---

### Official Review · AnonReviewer2 · 2019-04-25
**Preliminary ideas on an extremely useful PDDL extension**

**Rating:** 5
**Confidence:** 4

**Review:**

This paper proposes to extend PDDL with the keywords "new" and "delete"
which may appear in action effects and spawn or remove objects in the
planning task.

The (dis-)appearance of objects is a recurring problem in real-world
applications, and I have worked around this problem myself on several
occasions and therefore find an extension like this extremely useful.
Unfortunately, the work is still *very* preliminary and the idea is only
presented in the form of an example where a single object appears
without any dependence on existing or simultaneously created other
objects. Some additions and use cases I would have liked to see
discussed include (but are certainly not restricted to) the following:

1. An example I had to deal with recently was a Mars rover where objects
of interest (of type object-of-interest) can be detected whenever a
traverse action was executed. Each object of interest has an assigned
location (of type location), which is modeled with the predicate
object-at(object-of-interest, location). How would you expand a traverse
action between 2 locations to an action where both an object-of-interest
and a location are created such that the created object of interest is
at a newly created location? I assume the following would do the trick:


(:action traverse
  :parameters (?l1 ?l2 - location)
  :precondition (rover-at ?l1)
  : effect (and (not (rover-at ?l1))
                (rover-at ?l2)
                (new (?o - object-of-interest ?l - location)
                    (:init (object-at ?o ?l)))
           )
)

Is this something you planned to allow or will this cause issues I
oversee at the moment?

2. If objects are added, there must be some clever mechanism to set them
into relation with other objects that have already existed, and one
possibility are "conditional initializations". Take, for instance a
planning task with a predicate (pred ?o1 ?o2 - obj) that should hold for
a newly created object ?a of type obj and all existing objects ?b of
type obj if and only if another predicate (cond ?o - obj) holds for ?b.
Would you use the same syntax that is used for conditional effects,
i.e.,

(new (?a - object) (:init (forall (?b - object) (when (cond ?b) (pred ?a ?b)))))

or do you have something else in mind?

3. You indicate at least twice that action effects are only one option
to use the "new" and "delete" keywords. Where else do you think they can
be used?

4. You emphasize several times that the introduction of "new" and
"delete" will lead to states of variable size, but say nothing about a
variable size of ground actions. I believe this will be even harder on
the implementation side than a variable state size as concepts like
invariant synthesis or reachability analysis are significantly harder if
the number and form of actions is unclear.

5. Another point that should be addressed is the influence of added and
deleted objects on the goal. A question that immediately comes to mind
is what happens in an instance with:

(:objects o1 o2 o3 - my-object-type)
(:goal (and (achieved(o1) achieved(o2) achieved(o3))))

where the predicate "achieved" must be true for all (initially existing)
objects o1, o2 and o3. What happens now if o1 is deleted? Is
achieved(o1) removed from the goal along with it or is it no longer
possible to reach a goal state after o1 has been removed? I'd assume the
latter is more intuitive, because the object is explicitly mentioned in
the goal and it's impossible to achieve that goal once o1 has
disappeared. However, what happens if achieved(o1) was true in the
moment that o1 disappeared?

Furthermore, what happens if the goal is given via a quantifier as

(:objects o1 o2 o3 - my-object-type)
(:goal (forall (?o - my-object-type) (achieved(?o))))

which is equivalent under the current PDDL semantics. Intuitively, I
would+d assume now that removing o1 is not a dead end as it is still
possible to achieve the goal for all existing objects since o1 is not
mentioned explicitly, but then the two goals are no longer equivalent.

Similarly, what happens in the second case if another object of type
my-object-type is created?

6. I believe that your Theorem 1 should be correct once you have
specified the semantics of "new" and "delete" properly, but at the
moment it feels like you take the last step way before taking the
second. The whole proposed concept is too informal to be able to proof a
theorem at this point.


The second part on the paper is on a possibility to plan in this kind of
planning instance by using Maude. This part is very confusing and should
be improved significantly. It would be good if you would address the
following issues:

1. Introduce the planning formalism (or at least the parts you use)
properly. Of course, this is a workshop in a planning conference, but
people outside the community might be interested in this topic, and they
will certainly not be able to guess that operators consist of a
precondition and add and delete lists, so it wouldn't hurt to say this
at least once.

2. It's not correct that "the general assumption in PDDL is $D \subseteq
P$". In PDDL, there isn't even a delete list, but it's also not true if
PDDL is replaced with STRIPS. You could analyze if a compilation to
transition normal form (F. Pommerening and M. Helmert. "A Normal Form
for Classical Planning Tasks") preserves the task properties you require
here, though.

3. Try to introduce at least some concepts of Maude. Do you really
expect the reader to understand you example, e.g. the part
"next(x:Nat)"? This is far beyond the "Maude applies LHS => RHS rules"
explanation of Maude and it would be good to get this explained properly
if it is important.

4. Why is it even necessary to use Maude? Breadth-first search should
also be able to provide a solution to this kind of planning task and is
a concept that is universally understood by people attending the
workshop. Please help me understand the necessity of this part of the
paper and why it is relevant to the proposed PDDL extension.

5. What do you have in mind with the parts on model checking and the
graph transformations? I have to admit that I had already given up when
I reached this part of the paper and I might miss an obvious point, but
please be so kind and let me know why this is relevant for the paper and
what you plan to do to make sure a reader understands the connection
without having to work through a pile of papers and books.


To summarize, I believe this is a great topic and I would like to see
PDDL extended in this way. However, I am not certain if the paper isn't
too preliminary even for a workshop publication.


Minor comments:
- There are many typos and grammar errors and I'm not going to list all
  of them. Please make sure to put some more effort into the write-up.

- Why do you use "<name>" in the syntax specification of new instead of
  "<type>"? This is very confusing.

- Your pick-new action in Figure 1 should have a precondition "not
  arm-empty" and the caption mentions a "create block" action that does
  not exist

- You can extend your compilation to also cover the delete case by
simply adding an additional "not is-deleted" precondition to all
actions.